# Factors Influencing Implementation, Sustainability and Scalability of Healthy Food Retail Interventions: A Systematic Review of Reviews

**DOI:** 10.3390/nu14020294

**Published:** 2022-01-11

**Authors:** Adyya Gupta, Laura Alston, Cindy Needham, Ella Robinson, Josephine Marshall, Tara Boelsen-Robinson, Miranda R. Blake, Catherine E. Huggins, Anna Peeters

**Affiliations:** 1Global Obesity Centre, Institute for Health Transformation, School of Health and Social Development, Faculty of Health, Deakin University, Geelong, VIC 3220, Australia; laura.alston@deakin.edu.au (L.A.); cindy.needham@deakin.edu.au (C.N.); ella.robinson@deakin.edu.au (E.R.); Josephine.marshall@deakin.edu.au (J.M.); tara.b@deakin.edu.au (T.B.-R.); miranda.blake@deakin.edu.au (M.R.B.); kate.huggins@deakin.edu.au (C.E.H.); anna.peeters@deakin.edu.au (A.P.); 2Deakin Rural Health, Faculty of Health, Deakin University, Geelong, VIC 3220, Australia; 3Research Unit, Colac Area Health, Colac, VIC 3250, Australia

**Keywords:** healthy retail, choice architecture, marketing mix, food environment

## Abstract

The aim of this systematic review of reviews was to synthesise the evidence on factors influencing the implementation, sustainability and scalability of food retail interventions to improve the healthiness of food purchased by consumers. A search strategy to identify reviews published up until June 2020 was applied to four databases. The Risk of Bias in Systematic Review tool was used. Review findings were synthesised narratively using the socio-ecological model. A total of 25 reviews met the inclusion criteria. A number of factors influenced implementation; these included retailers’ and consumers’ knowledge and preferences regarding healthy food; establishing trust and relationships; perceived consumer demand for healthy food; profitability; store infrastructure; organizational support, including resources; and enabling policies that promote health. Few reviews reported on factors influencing sustainability or scalability of the interventions. While there is a large and rapidly growing body of evidence on factors influencing implementation of interventions, more work is needed to identify factors associated with their sustainability and scalability. These findings can be used to develop implementation strategies that consider the multiple levels of influence (individual, intrapersonal and environmental) to better support implementation of healthy food retail interventions.

## 1. Introduction

The food environment is “the physical presence of food that affects a person’s diet, a person’s proximity to food store locations, the distribution of food stores, food service, and any physical entity by which food may be obtained, or a connected system that allows access to food” [1]. Food retail settings are common settings where communities can purchase food on a day-to-day basis, such as supermarkets [2]. There is growing evidence to suggest that altering the food retail environment through interventions, such as changing product, price, promotion, and placement, are instrumental in promoting healthy choices and reducing the burden of non-communicable diseases [3]. In view of this, the World Health Organization also advises governments worldwide to “develop policy measures that engage food retailers and caterers to improve the availability, affordability, and acceptability of healthier food products [4]”.

Health-promoting interventions targeting food stores are shown to improve the healthiness of food purchased by consumers [5,6]. Recently, reviews have reported on the factors influencing implementation of food retail interventions with limited or no indication on the factors influencing sustainability and scalability of the interventions [7,8,9]. This is important information to ensure that factors that determine implementation (e.g., acceptability, engagement) align with sustainability (e.g., feasibility, organisational support) and scalability (e.g., integration with wider policy and in different contexts), and are integrated in the intervention development stage to maximise the sustained impact and scale-up of the interventions [10]. Specifically, there remains a significant gap in understanding on what factors influence sustainability and scalability of food retail interventions and if and how the factors influencing implementation, sustainability and scalability of food retail interventions interact with each other to influence the retailer’s decision-making. 

As sustained implementation and scale-up in practice requires actions from across the system (i.e., the context or set of contexts within which an intervention takes place) and different actors (including policymakers, practitioners, retailers and others) [11], it is critical to identify factors influencing sustainability and scalability across multiple levels (individual, intrapersonal and environmental) as well as explore if and how they interact with each other to impact retailer’s decisions to implement, sustain and scale-up food retail interventions to strengthen the system’s preparedness to accept the change and reduce resistance by stakeholders. There are numerous implementation frameworks and models that can support implementation, sustainability and scale-up of interventions, such as The Consolidated Framework for Implementation Research-CFIR [12]; Reach, Effectiveness, Adoption, Implementation, Maintenance-RE-AIM [13]; and the Theoretical Domains Framework-TDF [14]. However, these tools are useful for either determining the barriers or enablers to implementing a specific intervention or for evaluating dissemination success of an intervention but do not explain how factors influencing implementation, sustainability and scale-up of interventions span across the socio-ecological levels to establish sustained implementation success [15]. In view of this, we harnessed the strengths of the socio-ecological model [16] as this model explicitly helps with unpacking the complex interplay between multifaceted levels within a society and offers a greater understanding of how individuals and the environment interact within a social system that could become levers for changes to policy and practice. Identifying barriers and facilitators at multiple levels of the socio-ecological model is imperative to inform decision makers to implement healthy food retail initiatives that are likely to succeed and be sustainable at scale.

Thus, it is timely to synthesise evidence from across the existing reviews to build a more systematic understanding of factors influencing implementation, sustainability and scalability of the food retail interventions to inform healthy food retail initiatives that can be sustained over time and disseminated across different retail settings. To achieve this, we undertook a systematic review of reviews to synthesise the evidence on factors influencing implementation, sustainability (maintenance) and scalability of food retail interventions designed to improve the healthiness of food purchased by consumers.

## 2. Materials and Methods

A review protocol was developed *a priori* and registered in PROSPERO (The International Prospective Register of Systematic Reviews; registration number CRD42020190077) [17]. For reporting of this systematic review of reviews, the PRISMA (Preferred Reporting Items for Systematic Reviews and Meta-Analyses) guideline was followed [18].

### 2.1. Search Strategy

A modified PICOS (population, intervention/exposure, comparison, outcome and study context) criterion was developed (Table 1). Briefly, reviews that reported on factors influencing implementation, sustainability and scalability of interventions implemented by food retailer(s) that aim to improve the healthiness of food purchased by consumers and identified type of food retail outlets (except farmers markets or food pantries) and type of settings were included in this systematic review. Only reviews such as scoping reviews, systematic reviews and literature reviews were considered eligible for inclusion. Reviews summarising evidence from laboratory-based studies or modelling studies were excluded from the review. No exclusions were based on race, culture, ethnicity or geographical location of the food retail or retailers.

Utilising the expertise of the research team in food retail research and to undertake systematic reviews, a range of keywords and MeSH terms were identified to capture interventions implemented in food retail outlets and settings that aim to improve the healthiness of food purchased by consumers. The search was conducted in four databases: PubMed, Scopus, Embase and Web of Science. Database search terms were adapted from the following hedge terms: food retail outlets (type/settings), intervention foci and outcome measures. A full list of search terms under each hedge term can be found in Appendix A.

Using the same search terms, an online search using Google was also conducted to identify grey literature published up to June 2020 to expand the scope of the search. The screening process was initially limited to the first 100 uniform resource locators (URLs) depending on relevancy. Citation searches of included papers were performed (‘forward search’) and the reference lists of all included reviews and relevant review articles were searched to capture any citations missed by electronic searches (‘backwards search’). Search parameters were limited to only include review articles (scoping, systematic, narrative) published in the English language. All articles identified were subjected to selection criteria as described in the below section.

### 2.2. Study Selection

All reviews identified were imported into Covidence for screening. Following removal of duplicate articles, three authors (AG, LA and CN) independently screened titles and abstracts of the articles for their eligibility using PICO criteria (Table 1). Next, full texts were examined against the inclusion and exclusion criteria by the same three authors independently. Ten percent of these full texts were then cross-checked by two other authors (AG and AP). Any discrepancies between researchers were discussed to reach a final decision on articles for inclusion.

### 2.3. Data Extraction and Coding

Data were extracted from the results sections of all included reviews by three authors independently (AG, ER and JM) and results were compared by AG. All the eligible articles were reviewed, and the results were classified in two-steps. First, the results sections of each review were coded according to the elements described in Appendix A and reviews were classified into three categories to indicate if the review examined implementation, sustainability (maintenance) and scalability of food retail interventions. See Appendix A for definitions of the outcome measures. Any differences in implementation, sustainability and scalability of interventions across different types of retail outlets and in different settings were included, if reported in the review.

AG, ER and JM independently classified the reviews and AG cross-checked for consistency across all the reviews. Discrepancies were resolved through discussions between authors. Second, the factors classified under each category were further sub-classified into whether the factor is discussed as a barrier, a facilitator or both. 

### 2.4. Quality Appraisal

To assess credibility, relevance and value, each included review was critically appraised independently by three authors (AG, ER and JM) and duplicate quality appraisals of one-third of the reviews were conducted by AG, using the Risk Of Bias In Systematic reviews (ROBIS) [19]. The tool was completed in two phases: (1) identify concerns with the review process and (2) judge risk of bias in the review. In phase one, the risk of bias was assessed across four domains-study eligibility criteria; identification and selection of reviews; data collection and study appraisal; and synthesis and findings. The level of risk of bias associated within each phase was ranked to interpret the overall risk of bias (referred to as study quality hereafter) as low, high or moderate.

### 2.5. Data Analysis

A narrative synthesis of the results was conducted. Taking an inductive thematic approach, the data extracted were analysed by two researchers (AG and AP) to identify common patterns and themes. An initial coding framework was generated by AG based on repeated themes, and AG and AP then met to discuss discrepancies in coding and reach consensus on final themes that emerged from the data. The socio-ecological model framework was used to guide the coding. The final codebook included seven overarching themes that were defined and were subsequently categorised under three domains: implementation, sustainability (maintenance) and scalability. To draw meaningful conclusions from this review, the factors categorised under implementation, sustainability (maintenance) and scalability of food retail interventions were then grouped according to the socio-ecological model [16]. This model focuses on the interrelationships between individual, interpersonal (including social and community networks) and environmental (including organisation and policy) level factors [16]. The socio-ecological model is useful for better understanding the multiple factors that may determine a retailer’s decision making for implementing healthy food retail environment initiatives. This model allowed description of multifaceted factors broadly categorised under individual, interpersonal and environmental levels influencing the implementation, sustainability and scalability of food retail interventions.

## 3. Results

A total of 8879 published peer-reviewed review articles were identified from database searches. Following removal of duplicates (*n* = 2572), title and abstract screening was conducted for 6307 articles. Of these, 107 articles underwent full text review. Following full-text screening, 82 articles were excluded based on reasons listed in the PRISMA flowchart (Figure 1). The remaining 25 review articles were considered eligible for inclusion in the review. No grey literature met the eligibility criteria and as such none were included.

### 3.1. Review Characteristics

The 25 reviews were published between 2004 and 2020 with the majority published between 2016 and 2019 (*n* = 16). Of these, 20 were systematic reviews and five were narrative literature reviews. The characteristics of the included reviews are summarised in Table 2. The number of studies included in the reviews varied from 10 to 125 studies conducted across The Organisation for Economic Co-operation and Development (OECD) member countries; two narrative literature reviews did not report the number of studies included and the country in which the studies were conducted. Reviews included a range of stakeholders, including consumers, store managers, retail owners, food service business staff and management. No further sample demographic details were reported in any of the reviews.

### 3.2. Intervention Components

Interventions were identified across a variety of food retail outlet types (grocery stores, food stores, cafés, restaurants and vending machines) and settings (community/public settings, schools, universities, workplace and hospitals). Nearly all the reviews focused on assessing interventions based on marketing-mix and choice architecture strategies, including healthy defaults, priming or prompting; proximity strategies, including pricing, placement and promotion (at point-of-purchase and through media and advertising) [19]; and accreditation schemes to improve the healthiness of the food purchased. The duration of these interventions was reported in half of the reviews (*n* = 13) and varied from a single shopping trip to three years, while 12 reviews did not report the duration for which the intervention was delivered. The interventions were targeted both at healthy (e.g., water, fruits and vegetables) and unhealthy food (e.g., food high in risk nutrients that are non-essential and energy-dense) with the intention to promote uptake or increase the purchases of healthy foods and/or decrease the purchases of unhealthy foods.

### 3.3. Outcomes

Inconsistent measures were used to report on implementation, sustainability and scalability of interventions across reviews. All included reviews (*n* = 25) reported on measures such as retailer’s engagement, acceptability and adoption of food retail interventions to assess retailer’s ability and willingness to implement interventions in food retail outlets (*n* = 25). Some of these reviews, though less widely assessed, have also stated measures such as feasibility, retention and fidelity or cost-benefit and cost effectiveness to capture sustainability (*n* = 12) or scalability (*n* = 8) of interventions, respectively.

### 3.4. Quality of Included Reviews

The overall quality ratings of the included reviews using the ROBIS tool are stated in Appendix A. Spread across the categories of implementation, sustainability and scalability, 12 reviews were rated as high quality, two reviews were rated as moderate quality and the remaining 11 reviews were rated as low quality. While reviews had pre-defined eligibility criteria and presented a clear synthesis of the study findings, several reviews did not provide sufficient information on their study identification and selection process and did not describe any efforts undertaken to reduce bias in data collection and quality appraisal processes.

### 3.5. Overview of Reviews

While nearly all of the reviews reported on factors influencing implementation, only some reviews reported on factors influencing sustainability [7,9,20,21,22,23,24,25,26] and scalability [7,9,20,21,22,23,24,25,26] of healthy food retail interventions. Limited evidence of interactions among factors across multiple levels of the socio-ecological model that influenced retailers’ decision making were observed and are referred to throughout the results. All evidence is summarised in Table 2. To present a reasonable level of detail extracted from the included reviews, findings only from high quality reviews are summarised in the narrative synthesis; however, in case of insufficient high-quality reviews, findings from moderate and low-quality reviews were also included. To gain a more comprehensive understanding on the factors influencing implementation, sustainability or scalability of the interventions, the results were synthesised using the socio-ecological model (Figure 2).

### 3.6. Individual Level Factors

Retailer knowledge, skills and preferences regarding healthy food (and interventions) (*n* = 9):

Nine reviews [7,8,9,20,22,23,25,27,28] of mixed quality reported that retailer knowledge, attitudes and beliefs about healthy food and healthy food interventions are important factors that determine their ability to implement and sustain interventions.

Implementation and sustainability (maintenance): A recent high-quality mixed methods (i.e., qualitative and quantitative) systematic review [8] identified studies that reported retailers’ knowledge and beliefs as both barriers and facilitators to implementation of menu labelling interventions, using an existing conceptual implementation framework. A moderate-quality literature review [28] that mapped the influencers on food choice in remote Indigenous communities in Australia reported that store managers’ attitudes and beliefs towards food in such communities can influence the range of products stocked and their stock management practices, which can further influence choice, quality and availability of healthy food. For example, perceived lack of consumer “demand” was used as justification for not stocking certain healthy foods by a manager in a remote community as he perceived the foods were “unpopular”. Similar findings were also reported in two high-quality systematic reviews [9,20] where retailers considered the food-store intervention too complex for implementation due to lack of required knowledge, and its perceived low relevance, leading to a low level of satisfaction with the intervention strategy. No differences in factors were identified by type of food retail outlets and settings in which the retail outlets operated. Another high-quality systematic review reported that food store retailers’ perceptions around training designed to enhance retailer aptitude to deliver and sustain interventions negatively impacted their decision making and willingness or ability to support a healthy food and beverage intervention [7]. In two other (high [27] and low [22] quality) reviews, storeowners’ attitude and level of cooperation for intervention success resulting from incentives, both monetary and material support, and cultural and ethnic considerations were identified as critical factors in storeowners’ motivation to implement healthy food interventions.

Scalability: No review indicated any individual level factor that may impact food retailer’s ability to scale-up a food retail intervention.

### 3.7. Interpersonal Level Factors

#### 3.7.1. Consumer Preferences, Trust, Relationships and Demands (*n* = 11)

Eleven mixed quality reviews [5,7,8,9,20,23,25,27,28,29,30] reported that building trust and relationships between retailers and consumers, taking into consideration consumer preferences and demands, positively informed the retailer’s ability to implement and sustain interventions.

Implementation and sustainability: A high-quality systematic review [20] reported consumer perception (consumer level of satisfaction with the healthy food retail strategy, consumer store satisfaction) and business outcome (economic impact) perspective as a precursor to successful implementation and sustainability of healthy food retail strategy. The central role of consumer engagement with in-store interventions (for example, cooking demonstrations/taste tests and interactive education) was identified as another key element for long-term success of interventions [27]. Establishing consumer trust, offering good consumer service and consumer taste preference were identified as important elements that facilitated or impeded the implementation success and sustainability of food-store interventions as reported in one high-quality systematic review [7]. One high quality review stated that ‘consumers have the power to influence what is being sold in food outlets through their demand; and, if interventions can convince consumers to choose healthy foods, sustainability of the interventions is more likely’ [27]. A second high quality review [7] also highlighted that consumer preferences often influenced retailers’ strategies for stocking healthy products that resulted in increased consumer demand. Low popularity of and reduced demand for healthy foods among consumers, and consumers uninterested in health, were reported as barriers to food retailer’s decision-making process in several high [9,25], low [23,29] and moderate [28] quality reviews. A recent high-quality mixed method review [8] identified consumer needs and preferences (such as lack of consumer demand for/interest in menu labelling) as both a barrier and facilitator to implementation of menu labelling interventions.

Scalability: Only one high-quality Cochrane review [25] highlighted that consumer discontent (low popularity of and reduced demand) for healthy foods were reported as barriers to food retailers’ ability to scale-up in-store changes promoting healthy foods.

#### 3.7.2. Establishing Partnerships (*n* = 8)

Eight mixed quality reviews [7,8,9,22,23,30,31,32] included evidence that establishing partnerships with a range of stakeholders (health professionals, food suppliers and producers, and public and private sector organizations) inform retailers’ ability to deliver effective and sustainable interventions.

Implementation and Sustainability: A high-quality systematic review [7] of factors that influence food store owners’ and managers’ decision making to encourage healthy consumer purchases found establishing relationships between interventionists, retailers and staff was important for the success of food-store interventions. The review reported that trust and partnerships between retailer and interventionist was imperative to ensuring that proposed interventions fit the needs and resources of the system (e.g., are not in competition with policy or profit). The review also reported that similar socio-cultural backgrounds of retailers and intervention/research personnel were perceived as beneficial for establishing partnerships and trust. The review highlighted that food outlet retailers are key intermediaries between researchers and consumers and staff. Engaging food outlet retailers in dissemination practices to keep them informed and involved throughout the entire process of intervention development was seen as critical to facilitate intervention implementation and enhance sustainability. Another high-quality systematic review [9] on healthy food outlet interventions noted that in about 12% of their total studies engagement (with retailers through retailer-specific engagement strategies such as recurring contact and providing staff training) and collaboration (between interventionist and retailer) are key contributors to successful implementation and sustainability of interventions. Studies in the review identified that co-creation of the intervention was helpful to manage contextual barriers, improve the context–intervention fit, and foster feelings of ownership among the retailers. Similarly, in a high-quality Cochrane review [25], evidence from stakeholder interviews showed that strong executive support was crucial for sustained implementation of interventions. A recent high-quality systematic review [8] found studies identifying lack of engagement with stakeholders as a barrier for retailers’ decision-making ability to adopt/implement menu-labelling interventions.

Scalability: No review indicated any evidence that establishing partnerships may impact a food retailer’s ability to scale-up food retail interventions.

### 3.8. Environmental Level Factors

Environmental level factors reported included profitability, organisational support (control and ownership over food store supplies) and resources (staff, time, capital), in-store infrastructure (store format and location) and health-promoting polices influenced a retailer’s ability to implement, sustain and up-scale interventions in food retail settings.

#### 3.8.1. Profitability (*n* = 15)

Fifteen mixed quality reviews [7,8,9,21,22,23,25,26,27,29,30,33,34,35,36] reported that increased sales, improved revenues or total profits facilitated retailer’s ability to implement, sustain and scale-up health promoting interventions in retail settings.

Implementation and sustainability: Several reviews highlighted that no loss in profit motivated the retailers to implement, sustain and scale-up the interventions (high [26,27,33,36]; low [22,23,30] quality reviews). A high-quality Cochrane review [25] reported concerns among retailers about long term impact of intervention on business due to lack of profitability. Another recent high-quality systematic review [7] concluded that slim profit margin negatively impacts food store owner and manager’s ability or willingness to implement and sustain strategies to encourage healthy consumer purchases across various food retail outlets. A recent high-quality systematic review [8] concluded that implementation of menu labelling interventions by the retailer is negatively influenced by reduced sales or profitability. Similar findings were also reported in another high-quality systematic review [9] where commercial interest (in the form of profit) was described as a major determinant in retailers’ organisational decision making and conflict between commercial interests and intervention interests sometimes presented as a barrier for retailers’ decision-making.

Scalability: Eight reviews [7,8,9,20,21,22,23,26] highlighted that increased total profits facilitated a retailer’s ability to scale-up health promoting interventions in retail settings.

#### 3.8.2. Organisational Support (Control and Ownership over Food Store Supplies) and Resources (Staff, Time, Capital) (*n* = 10)

Ten mixed quality reviews identified lack of organisational support in the form of ownership and insufficient resources limited retailer ability to implement, sustain and scale-up food retail interventions [7,8,9,20,23,24,25,29,37,38].

Implementation and sustainability: A large high-quality systematic review [7] examining food environments described that retailers often noted limited control over the foods and beverages available in food stores limiting their ability to implement healthy interventions. In another high-quality systematic review [9], organisational factors, such as thorough planning, structure and transparent decision making, were sometimes named as facilitators to a retailer’s ability or willingness to offer healthier options. Similar factors were also reported in two other high [8,25] and one low quality review [38]. A large high-quality systematic review [7] examining food environments reported factors such as long working hours, managing work outside their job description, high employee turnover rate, costly and difficult dynamics of coordinating a business and competition with other food stores to generate revenue, as barriers to a retailer’s ability or willingness to offer healthier options. In another high-quality systematic review [9], the intervention process was perceived to be constrained by shortcomings in task management, planning or limited time and staff turnover. Similar barriers were also reported in two other high-quality systematic reviews [8,25].

Scalability: No review reported any evidence to understand if organisational support (control and ownership over food store supplies) and resources (staff, time, capital) may impact a food retailer’s ability to scale-up food retail interventions.

#### 3.8.3. In-Store Infrastructure (Store Format and Location) (*n* = 6)

Six mixed quality reviews identified store infrastructure in the form of store format and store location as factors influencing a retailer’s ability to implement, sustain and up-scale food retail interventions [5,7,8,9,22,25].

Implementation and sustainability: A large high-quality systematic review [7] examining food environments identified the convenience store format, such as displaying quick-grab items rather than grocery products, was more preferable to the retailers due to high consumer demand and that it conflicted with their healthy food goals, limiting the retailer’s ability to implement healthy food retail interventions. The review also identified that clean and well-structured food store environments were important for consumer shopping experience and that the latter enabled retailers to implement health-promoting strategies, such as altering food store stocking practices, profile, promotion or pricing strategies. Enhancing the availability of perishable products was a concern for retailers due to low consumer demand and thus limited the retailer’s ability to stock products. The review also described food store location as both beneficial (when located in a dense residential area with minimal competition) and detrimental (when located in rural areas that impacted sale) to retailers for implementing interventions. In another high-quality systematic review [9], food store structure (physical and operational) were described as important barriers to a retailer’s ability or willingness to offer healthier options. Physical structures were most frequently discussed in the form of space constraints; limited storage/cooling facilities and store renovations. Operational structural barriers included contractual obligations regarding product placement and stocking. Organisational factors, such as thorough planning, structure and transparent decision making, were sometimes named as facilitators. Similar structural barriers were also reported in two other high-quality systematic reviews [8,25].

Scalability: One high-quality systematic review [9] described food store structure (physical and operational) as important barriers to a retailer’s ability to scale-up food retail interventions offering healthier options. Physical structures were most frequently discussed in the form of space constraints; limited storage/cooling facilities and store renovations. Operational structural barriers included contractual obligations regarding product placement and stocking.

#### 3.8.4. Enabling Policies (Local and Federal Level) That Promote Health (*n* = 5)

Five reviews [7,28,30,39,40] reported on how health promoting policies may influence the ability of retailers to implement healthy food retail interventions.

Implementation and sustainability: In a high-quality systematic review [7], both local and federal policies influenced food retailers’ decision making. Local policies prohibited retailers from utilising nearby agriculture avenues to support healthy food stocking practices, while federal policies such as the Supplemental Nutrition Assistance Program (SNAP) and Women, Infants, and Children (WIC) food package changes were described to positively impact profits and ensured consumer demand, enhancing retailers’ ability to offer diversity in stocking healthy products.

Scalability: Only one high-quality systematic review [7] identified that presence of local and federal policies was a barrier and also a facilitator influencing food retailers’ ability to scale-up food retail interventions.

## 4. Discussion

Food retailers are integral to successful adoption, maintenance and dissemination of interventions promoting healthy food retail in practice. This systematic review of reviews brings together for the first time the available evidence on factors affecting store-based interventions. This review demonstrates (1) there is a detailed evidence-base of factors affecting implementation of interventions across each of the levels of environmental, interpersonal and individual; (2) there is limited evidence to understand what factors contribute to sustainability and scale-up of interventions, and this has been consolidated here to strengthen understanding and guide future research. Overall, the evidence shows that implementation and to some extent, sustainability of health promoting food retail interventions, are more likely to be successful in practice if they are aligned across multiple levels of the socio-ecological model. Below, we discuss key results within each level of the socio-ecological model [16], each with research, practice and policy implications.

Among the *individual-level factors*, lack of retailer knowledge, skills and preferences regarding healthy food interventions was described as a key barrier in the food retailer’s ability to implement and sustain food retail interventions. This indicates that training regarding key elements of a healthy food environment, relevant to the food retailer to support them with knowledge and offer practical strategies, has the potential to benefit implementation. A further potential strategy emerging from this review is providing information to retailers on how interventions may create both value for consumers [41] as well as maintain profit for themselves and their stakeholders [42]. This is in line with concepts of business theory and organisational management, which argue that addressing societal needs and stakeholder interests is key to business success and profitability. Previous studies have shown that providing ongoing business training and technical assistance (offered by trained professionals) to retailers may help raise awareness and enhance skills and preferences of retailers to stock a range of healthy products [43,44]. Training materials, such as toolkits [45], could be a useful resource in the training of food retailers on ways to create healthy recipes and engagement strategies. However, given the diversity in food retail outlets and settings in which the food retail outlet operates; and inconclusiveness of whether the enablers and barriers may differ between each setting, by using the evidence generated from this review we can now engage with retailers to co-design intervention that are context-specific to inform the feasibility of the scale-up of food retail interventions.

At the *interpersonal level*, retailer trust and partnerships with consumers and a range of stakeholders, such as health professionals, food suppliers and producers, public and private sector organisations, were identified as two strong themes that facilitated retailers’ ability to deliver effective and sustainable interventions. Engaging with communities and stakeholders from inception to completion of the projects was identified as fundamental to building trust [7]. Recent evidence [46] also suggests that co-designing food retail interventions reduces contextual barriers, improves the context-intervention fit, and stimulates feelings of ownership among consumers, retailers, and other stakeholders. Our review shows this is a gap that needs to be filled to offer enhanced support to the retailers. Furthermore, working with community members to tailor interventions or develop culturally relevant interventions according to their needs, preferences and choices makes co-design a promising approach for implementation, sustainability and scalability of interventions [47,48,49]. One way to build strong trust and partnership between retailers, communities and stakeholders is by enabling retailers to adopt various engagement strategies, such as keeping recurring contact with stakeholders and consumers, providing staff training, offering good customer service and building multi-stakeholder collaboration [50]. Future studies should also explore the potential of evidence-based co-design approaches [46] to improve the success of healthy food retail intervention implementation, sustainability and scalability.

At the *environmental level*, four factors, namely, profitability, organisational support and resources, in-store infrastructure and enabling health promoting policies emerged as factors strongly influencing retailer’s decision-making towards implementing healthy food retail interventions. Limited space and resources inhibited retailers’ ability to implement healthy food retail interventions. These barriers were reported to be often exacerbated by power imbalances between retailer and supplier. An indirect consequence of this power imbalance results in the prioritising of profit over health. Future research is needed to identify ways to rescale this power imbalance in favour of health. Retailers’ commercial interests, particularly a profit-driven approach, was identified as a key determinant affecting the implementation of interventions. Identifying and implementing interventions that do not compete with the business’s profit margins can enhance implementation and sustainability. This can be achieved through systems change endeavours, such as the policies, regulations, relationships, resources, power structures and values [51] that could offer retailers with a level playing field to promote healthy choices without losing their market share. For example, there was some indication in our review, albeit limited, that environmental level factors such as policies promoting ownership and participation may offer potential opportunities for retailers to implement healthy interventions—reflecting the principle of strategic corporate social responsibility [52].

There was limited evidence of interactions among factors across multiple levels of the socio-ecological model that influenced retailer decision making. For example, factors such as managers’ *attitudes and beliefs* towards food (at individual level), consumer demand and engagement and collaboration with stakeholders (at interpersonal level) influenced retailers’ stock management practices including their decision for implementing and sustaining stocking healthy products or implementing in-store interventions such as menu labelling (at environmental level). Further, having similar socio-cultural backgrounds of retailers and research personnel were reported to promote partnerships and trust between the two, indicating some interactions between the factors operating at individual level and interpersonal levels of the socio-ecological model. Future investigations in unpacking these interrelationships across multiple levels of the socio-ecological model would be useful to support retailers to implement successful healthy food retail initiatives that are also sustainable at scale.

While most of the evidence in the included reviews focussed on detailing the factors influencing the food retailers’ ability to implement food retail interventions, a very small number of reviews offered some indication of factors influencing sustainability of food retail interventions and a much smaller on the factors influencing scale-up of food retail interventions. For a true public health impact of the food-retail interventions, there is a need for evidence on how retailers can be enabled to maintain and scale-up the interventions to reap health and cost-related benefit over time. Through this review we have highlighted some key information on factors that influence sustainability and scalability that can be built upon in future studies on sustainability and scale up. This is important as without this information, researchers, practitioners and decision makers lack the ability to maximize health and financial impact of available effective food retail interventions and polices. By consolidating the evidence in this systematic review of reviews, we may now be better positioned to test multi-component interventions to overcome barriers to adoption, sustainability and scale up. Planning and developing all aspects of implementation, sustainability and scalability through a co-design process [48,49] with multiple stakeholders is essential to achieve success and for decision makers to replicate the evidence-based strategies into practice that can benefit both the retailers and consumers. These gaps in evidence indicate a significant lack in evaluation of food-retail interventions beyond the implementation stage.

Employing a socio-ecological model enabled a greater understanding of the factors that influence implementation, sustainability and scalability across multiple levels to inform retailer’s ability to implement, sustain and scale-up of food retail interventions. Future research applying implementation frameworks could use the findings from this review to design and examine intervention strategies across the socio-ecological levels to achieve implementation success and beyond. Our review highlights the need to embed sustainability and scalability as key concepts within the implementation research to facilitate translation of evidence into the real world and ensure long term and widespread benefits of the interventions. Further, exploring the above dimensions of implementation, sustainability and scalability in different settings and contexts (such as different retail settings, rural vs urban areas, different socio-economic groups, low-middle income countries) is a critical next step to enhance adaptability of interventions in the most acceptable way.

### Strengths and Limitations

Strengths and limitations *of this review of reviews* include a robust search strategy that was developed and adapted for four large databases to ensure no relevant reviews (both academic and grey literature) were missed. Second, all the screening processes were conducted in duplicate by three authors independently and a further 10% were cross-checked by two other co-authors to ensure accuracy in the selection process. Third, a well-defined study selection criterion and independent coding of the findings to ensure the accuracy of the findings made this review process rigorous and robust. Fourth, findings were summarised using a socio-ecological model [16] to provide a comprehensive understanding of the topic. Among the limitations, only reviews published in English language were included in this systematic review, which may have led to exclusion of relevant trials and evaluation studies not captured within the reviews included, particularly recently published studies. Second, it is possible that similar studies may have been included in more than one review and this may have led to some heterogeneity in the findings within the individual reviews as well as our review.

There are also some strengths and limitations *of the reviews* included in this review. Among the strengths, nearly half the reviews were of moderate to high quality as assessed through the quality assessments. Further, the reviews provide a large volume of insightful evidence (~800 studies) across multiple food outlet types and settings to assist with an extensive narrative synthesis. However, there was high heterogeneity in the data collection tools and reporting of outcome measures within the included reviews. Furthermore, the reporting of the interaction between the factors influencing implementation, sustainability and scalability of interventions in the majority of the studies was limited. Last, few studies discussed the factors influencing sustainability and scalability of food retail interventions.

## 5. Conclusions

This systematic review of reviews provides comprehensive evidence suggesting that implementation of interventions aimed at making food retail environments health promoting require targeting a combination of individual, intrapersonal and environmental factors. Consideration to how contextual factors may be linked to retailers’ perceptions is necessary to increase the likelihood of sustained implementation and for potential scale up. The overall findings of this review will support researchers and retailers to develop, tailor and test strategies to address barriers and leverage facilitators that assist with implementation, sustainability and scalability of healthy food retail interventions. The findings of this review can be used as a starting point to help build research-practice partnerships that support business and health. Future work focusing on evaluation of food-retail interventions beyond the initial implementation phase is warranted to support sustainability and scalability of these interventions.

## Figures and Tables

**Figure 1 nutrients-14-00294-f001:**
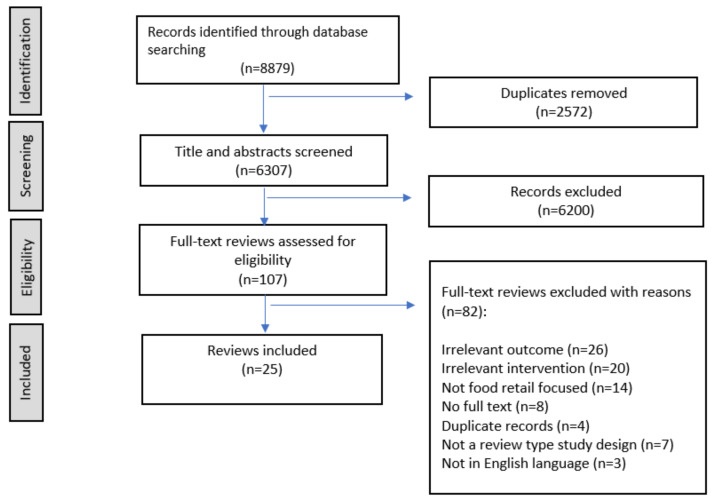
PRISMA flowchart.

**Figure 2 nutrients-14-00294-f002:**
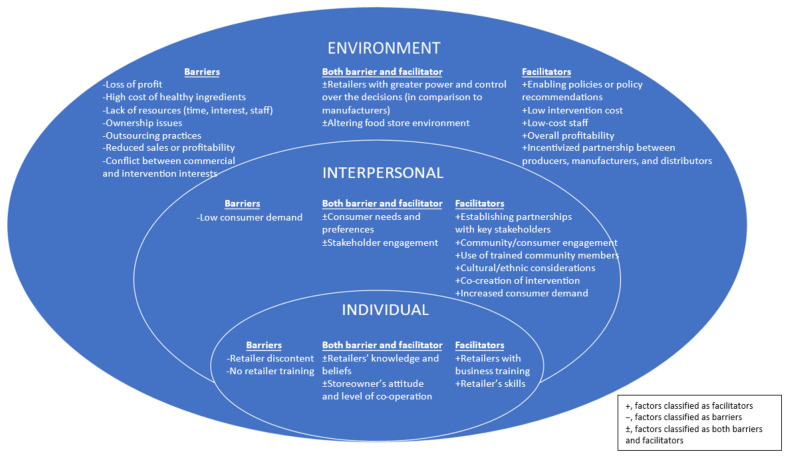
Modifiable factors influencing implementation, sustainability and scalability of food retail interventions from all included reviews (*n* = 25) using the socio-ecological model.

**Table 1 nutrients-14-00294-t001:** PICOS criteria.

Criteria	Inclusion	Exclusion
Population	Studies that included and identified: type of food retail outlets (defined as an establishment primarily engaged in retailing a general line of food, e.g., cafeteria, grocery store); type of settings (e.g., food retail outlets located in hospital, university, school)	Food retail outlet such as farmers markets or food pantries. No exclusions were based on race, culture, ethnicity or geographical location of the food retail or retailers.
Intervention/exposure	Studies that aim to improve the healthiness of food purchased by consumers by altering factors such as price, promotion, placement	Studies that do not include a relevant intervention.
Comparator	No restrictions	
Outcome	Studies reporting on implementation, sustainability and scalability of interventions implemented by food retailer(s) that aim to improve the healthiness of food purchased by consumers (see definitions in Appendix A)	Studies that do not report a relevant outcome.
Type of Studies	Review studies (scoping, systematic, literature); English language published from inception to June 2020	All studies except reviews. Reviews summarising evidence from lab based or modelling studies.

**Table 2 nutrients-14-00294-t002:** Characteristics and findings of the included reviews (*n* = 25).

Author/Year of Publication; Type of Review; Number of Studies in Review (*N*)	Aim of the Review	Type of Food Retail Outlet	Type of Setting	Type of Intervention	Factors Influencing Implementation of Intervention	Factors Influencing Sustainability of Intervention	Factors Influencing Scalability of Intervention	Risk of Bias
Adam and Jensen 2016 [20] Systematic review *N* = 42	To systematically review the literature on effectiveness of food store interventions intended to promote the consumption of healthy foods	Grocery stores, supermarkets, and convenience stores	Community-based setting	Affordability (price), information and access/availability	± Storeowner’s attitude and level of co-operation - Conflicts among intervention partners + Use of trained community members - Financial losses (perceived or actual) due to intervention; + Cultural/ethnic considerations + No profit loss	± Storeowner’s attitude and level of co-operation + Community/consumer engagement + No profit loss	Not reported	High quality
Beltran and Romero, 2019 [21] Systematic review *N* = 20	To identify relevant topics in the literature about healthy eating and restaurants	Restaurants	Community-based setting	Point of sale information, events and sales promotions	- Low consumer demand - Anticipated harm to profit - Higher cost of healthy ingredients - Concerns around loss of profit	Not reported	Not reported	Low quality
Blake et al., 2019 [22] Systematic scoping review *N* = 107	To synthesise business outcomes of healthy food retail initiatives	Food and beverage stores, restaurants, vending machines	Community-based setting (except schools)	Availability, price, place, promotion	- Lack of required knowledge - Perceived low relevance leading to low level of satisfaction with the intervention strategy + Consumer satisfaction	+ Profit driven	+ Profit driven	High quality
Bucher et al., 2016 [23] Systematic review *N* = 15	To investigate the effect of positional changes of food placement on food choice	Laboratory	University, hospital setting	Product placement (proximity or order)	± Retailers with greater power and control over the decisions (in comparison to manufacturers)	Not reported	Not reported	High quality
Buttriss et al., 2004 [24] Narrative review *N* = Not reported	To review factors that influence food choice	Cafeteria and restaurants	Primary care, universities, schools, workplace	Pricing strategies	+ Establishing partnerships with key stakeholders + Overall profitability + Adopting ‘a whole school approach’ policy	+ Establishing partnerships with key stakeholders	Not reported	Low quality
Cameron et al., 2016 [25] Systematic review *N* = 50	To determine the effectiveness of supermarket-based interventions on the healthiness of consumer purchases	Supermarkets	Community-based setting	Product promotion, placement, mass media interventions	+ Low-cost interventions that require little retailer input	+ Low-cost interventions that require little retailer input	+ Economically profit driven impact of the intervention on the retailer	Low quality
Escaron et al., 2013 [5] Systematic review *N* = 58	To synthesise the evidence on supermarket and grocery store interventions to promote healthful food choices	Supermarket and grocery store	Community-based setting	Point-of-purchase information (use of demonstrations, taste testing, signs, labels, printed materials)	+ Working with community members to develop culturally relevant interventions + Combining demand-and supply-side strategies	+ Working with community members to develop culturally relevant interventions	Not reported	Low quality
Gittelsohn et al., 2012 [26] Systematic review *N* = 16	To determine the impact of small-store interventions on food availability, dietary behaviors, and chronic disease risk	Small food store/corner stones, convenience stores, bodegas/tiendas and liquor stores	Rural and urban settings (in 6 countries)	Point-of-purchase (shelf labels, posters, coupons, vouchers, educational flyers, giveaways)	+ Store owners and staff education and business training + No profit loss	+ Incentivized partnership between producers, manufacturers, and distributors + No profit loss	+ No profit loss	Low quality
Gittelsohn et al., 2013 [27] Systematic review *N* = 19	To systematically review community-based interventions aimed to increase access to and consumption of healthful foods	Carry out, fast-food and restaurants	Community-based setting	Increase access to and consumption of healthy foods	- Perception of intervention as burdensome by food-source owners + Incentives (such as free menu analyses and point-of-purchase materials)	- Open volunteer enrollment leading to low reach (as assessed by counting consumers) + Engagement with staff	Not reported	Low quality
Gittelsohn et al., 2017 [28] Systematic review *N* = 30	To determine the effect of food-pricing interventions on retail sales, consumer purchasing and consumption of food	Grocery stores, supermarkets, farmers markets, cafeterias, restaurants, corner stores	Worksite, sports gym, school, swimming pool, hospitals	Pricing intervention (alone or in combination with stocking, sales)	+ No profit loss	+ No profit loss	Not reported	High quality
Glanz et al., 2012 [29] Integrative review *N* = 125	To review research on in-store food marketing interventions	Grocery stores	Community-based setting	In-store food marketing (product, price, place, and promotion)	+ Greater retailer power and control over the decisions (in comparison to manufacturers)	Not reported	Not reported	Low quality
Grech and Allman-Farinelli., 2015 [30] Systematic review *N* = 12	To determine the efficacy of nutrition interventions in vending machine in eliciting behavior change to improve diet quality	Vending machines	Worksites, universities, and school setting	Point-of-purchase promotion, nutrition policy, availability, pricing and behavioral programs	- Concerns around loss of profit due to price reductions or restrictions on availability of unhealthy choices	- Concerns around loss of profit due to price reductions or restrictions on availability of unhealthy choices	Not reported	Moderate quality
Henryks & Brimblecombe, 2016 [31] Narrative literature review *N* = not reported	To identify and map key influencers of food choice at the point-of-purchase in Australian Remote Indigenous Communities and identify gaps in knowledge	Food stores	Remote Indigenous communities	Point-of-purchase influences	+ Store managers’ attitudes and beliefs towards food - Low consumer demand + Policy with multiple strategies (income management in combination with the stores licensing programs)	+ Policy with multiple strategies (income management in combination with the community stores licensing programs)	Not reported	Moderate quality
Hillier-Brown et al., 2017 [32] Systematic mapping evidence synthesis *N* = 75	To identify and describe interventions to promote healthier ready-to-eat meals sold by specific food outlets	Food outlets selling ready-to-eat meals i.e., cafes, restaurants, quick service restaurants	Community-based setting (excludes schools, workplaces, institutions)	Heterogeneous, including award/accreditation and non-award, generally related to product and promotion	+ Project team’s skills, knowledge + Establishing relationships with staff and partnerships - Lack of time or interest + No profit loss	- Low consumer demand	+ No profit loss	Low quality
Hillier-Brown et al., 2017 [33] Systematic review *N* = 30	To systematically review the international literature on the impact of interventions to promote healthier ready-to-eat meals	Food service outlets	Community-based setting (excludes schools, workplaces, institutions)	Food reformulation, healthier offerings, accreditation scheme, price, labelling/information	+ Establishing relationships	Not reported	Not reported	High quality
Houghtaling et al., 2019 [7] Systematic review *N* = 31	To identify factors that affect food storeowner and manager decision making and ability or willingness to apply marketing-mix and choice-architecture strategies to encourage healthy consumer food and beverage purchases among consumers	Food store includes grocery or supermarket	Urban community-based setting	Place, profile, portion, pricing, promotion, healthy defaults, priming or prompting, and proximity	- No retailer training + Increased consumer demand + Trust and partnerships between retailer-interventionist ± Food store layout/location - Incomplete control of retailers over the foods and beverages available in food stores - Outsourcing practices - Long working hours, managing work outside their job description, high employee turnover rate, difficult to generate revenue - Slim profit margins ± Local and federal policies	- No retailer training + Increased consumer demand + Trust and partnerships between retailer-interventionist and consumers ± Food store layout/location ± Consumer service and consumer taste preference -Slim profit margins ± Local and federal policies	- Slim profit margins ± Local and federal policies	High quality
Hua & Ickovics, 2016 [34] Narrative literature review *N* = 10	To describe intervention designed to promote healthier vending purchases by consumers	Vending machines	Schools, universities, worksites, parks and buildings	Price, product availability, promotions/signage system, marketing, or education campaign	+ Profit due to price reductions	+ Profit made from price reductions	Not reported	Low quality
Kerins et al., 2020 [8] Mixed methods systematic review *N* = 17	To identify barriers and facilitators to implementation of menu labelling interventions from the perspective of the food service industry	Restaurants, food service corporations	Food service industry	Menu labelling format (numeric or interpretive), scheme (voluntary or mandatory) or type of food service business	± Retailers’ knowledge and beliefs ± Consumer needs and preferences ± Stakeholder engagement - Food store structure - Reduced sales or profitability	Not reported	- Reduced sales or profitability	High quality
Kraak et al., 2017 [35] Desk literature review *N* = 84	To evaluate restaurant-sector progress to create healthy food environments	Restaurants (chain and non-chain), includes quick-serve restaurants	Government, industry, non-governmental organizations, private foundations, academic institutions	Place, profile, portion, pricing, promotion, healthy default picks, prompting or and proximity	+ Comprehensive food and beverage marketing policies	Not reported	Not reported	Low quality
Liberato et al., 2014 [36] Systematic review *N* = 32	To review the effectiveness of interventions at point-of-sale to encourage purchase and/or eating of healthier food to improve health outcomes	Grocery stores, supermarkets, vending	Point of sale in any community-based setting	Infrastructure, monetary incentives, marketing strategies (promotion and placement)	+ No profit loss	Not reported	Not reported	High quality
Mah et al., 2019 [37] Systematic review *N* = 86	To update the evidence on the effectiveness of retail food environment interventions in influencing diet	Supermarkets, grocery stores, convenience stores, gas stations	Community-based settings (except schools, workplace setting)	Changing the availability or the product, pricing, placement, or promotion	+ Enabling policies or policy recommendations	Not reported	Not reported	Low quality
Marcano-Olivier et al., 2020 [38] Systematic review *N* = 25	To identify interventions using behavioral nudges to promote healthy food item choice or consumption	Cafeteria	School	Simple nudge-only interventions	± Altering food store environment	+Low-cost staff	+ Low intervention cost	High quality
Middel et al., 2019 [9] Systematic review *N* = 41	To identify barriers or facilitators to the implementation of healthy food-store interventions	Supermarket/food stores	Community/public setting	Any intervention that changes price, availability, promotion, or point-of-purchase information	- Lack of retailer’s knowledge - Perceived low relevance - Low consumer demand + Engagement and collaboration (between interventionist and retailer) + Co-creation of intervention - Food store structure - Conflict between commercial and intervention interests	+ Engagement and collaboration (between interventionist and retailer) + Co-creation of intervention - Food store structure - Conflict between commercial interests and intervention interests - Lack of profitability	- Food store structure - Conflict between commercial interests and intervention interests - Lack of profitability	High quality
von Philipsborn et al., 2019 [39] Systematic review *N* = 58	To assess the effects of environmental interventions on the consumption of unhealthy food and health outcomes	Cafeterias, canteens, kiosks, restaurants, convenience/grocery stores, supermarkets, vending machines	Schools, hospitals, leisure centers, theme park, workplaces	Labelling, nutrition standards, pricing, availability and promotion, food benefits, home-based interventions	- Stakeholder discontent e.g., consumer complaints and perceptions of the store - Low consumer demand - Food store structure (physical and operational)	- Food store structure (physical and operational) - Lack of profitability	- Stakeholder discontent e.g., consumer complaints and perceptions of the store	High quality
Wilson et al., 2016 [40] Systematic review *N* = 13	To investigate nudging interventions, and their effectiveness for influencing healthier choices	Canteen, cafeteria, fast-food restaurant	Universities, hospitals, self-service buffets	Visibility, accessibility, availability, labels (traffic light, calorie, descriptive), downsize meals, taste-testing	Not reported	+ No profit loss	+ No profit loss	High quality

+, factors classified as facilitators. -, factors classified as barriers. ±, factors classified as both barriers and facilitators. *N* = number of studies in reviews.

## Data Availability

Data are contained within the article or are available from the included studies that have been referenced throughout.

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
