# Peer review of "Factors Influencing Implementation, Sustainability and Scalability of Healthy Food Retail Interventions: A Systematic Review of Reviews"

_nutrients, 2022, doi:10.3390/nu14020294_

Round 1

Reviewer 1 Report

Thank you for allowing me to review this manuscript. This manuscript titled "Factors Influencing the Implementation, Sustainability, and Scalability of Health Food Retail Interventions: A Systematic Review 3 of Reviews". The objective of this systematic review of reviews was to synthesize the evidence on 14 factors that influence the implementation, sustainability and scalability of food retail interventions to improve the wholesomeness of foods purchased by consumers.
It is an interesting and highly relevant article today, although it has several limitations that make it susceptible to publication in this journal. These limitations are detailed below:

- The abstract should be a single paragraph and should follow the style of structured abstracts, but without titles: Background, methods, results and conclusion. Check this point to make this summary without titles.

- In the materials and methods section, it is described very adequately and in sufficient detail. However, it would be interesting to include the period of time necessary to write the article, as well as to clearly specify the inclusion and exclusion criteria that were taken into account.

- The tables do not specify a footer for the acronyms used in them.

- The conclusions are clear and precise and reflect the impact or importance of the results today. However, given the importance and topicality of the subject, it would be interesting to include some future lines.

- The bibliographic references should be reviewed. The year of the different sources consulted, following the regulations required by the magazine, is not in italics.

Author Response

  1. The abstract should be a single paragraph and should follow the style of structured abstracts, but without titles: Background, methods, results and conclusion. Check this point to make this summary without titles.

Author’s Response: Thank you for your suggestion. We have now removed the following titles: Background, methods, results and conclusion from the abstract following the journal’s guidelines.

  1. In the materials and methods section, it is described very adequately and in sufficient detail. However, it would be interesting to include the period of time necessary to write the article, as well as to clearly specify the inclusion and exclusion criteria that were taken into account.

Author’s Response: Thank you for your suggestion. We have stated that the search was conducted until June 2020 (Page 3, Lines 111). We have also now clearly specified our inclusion and exclusion criteria in the materials and methods section as, “Briefly, reviews that reported on factors influencing implementation, sustainability and scalability of interventions implemented by food retailer(s) that aim to improve the healthiness of food purchased by consumers and identified type of food retail outlets (except farmers markets or food pantries) and type of settings were included in this systematic review. Only reviews such as scoping reviews, systematic reviews and literature reviews were considered eligible for inclusion. Reviews summarising evidence from laboratory-based studies or modelling studies were excluded from the review. No exclusions were based on race, culture, ethnicity, or geographical location of the food retail or retailers” (Page 2-3, Lines 92-100)

  1. The tables do not specify a footer for the acronyms used in them.

Author’s Response: Thank you for your suggestion. We have now described the acronyms used (N = number of studies) as a footnote under Table 2. (Page 10, Line 196)

  1. The conclusions are clear and precise and reflect the impact or importance of the results today. However, given the importance and topicality of the subject, it would be interesting to include some future lines.

Author’s Response: Thank you for your comment. We have previously included several future implications of our review on Page 17, Lines 528-540. As suggested, we have now included a sentence in the conclusion as, “Future work focusing on evaluation of food-retail interventions beyond the initial implementation phase is warranted to support sustainability and scalability of these interventions”. (Page 18, Lines 576-578)

  1. The bibliographic references should be reviewed. The year of the different sources consulted, following the regulations required by the magazine, is not in italics.

Author’s Response: Thank you for your comment. We have now edited the bibliographic references to include accessed date for web records, amended author-names wherever abbreviated and also the year in italics. We have also thoroughly proofread the manuscript.

Reviewer 2 Report

In the manuscript "Factors influencing implementation, sustainability, and scalability of healthy food retail interventions: A systematic review of reviews" the Authors tried to synthesize the evidence on factors influencing implementation, sustainability and scalability of food retail interventions designed to improve the healthiness of food purchased by consumers. The advantage of the study is an important topic, and way of study, i.e. a systematic review of reviews. Moreover, the Authors used four databases. The review was registered in PROSPERO (no: CRD42020190077) and followed PRISMA guideline. The manuscript is written very clearly and strictly. All sections are described in a comprehensive manner. Methods are adequate. I am very impressed with the work done. 

Author Response

In the manuscript "Factors influencing implementation, sustainability, and scalability of healthy food retail interventions: A systematic review of reviews" the Authors tried to synthesize the evidence on factors influencing implementation, sustainability and scalability of food retail interventions designed to improve the healthiness of food purchased by consumers. The advantage of the study is an important topic, and way of study, i.e. a systematic review of reviews. Moreover, the Authors used four databases. The review was registered in PROSPERO (no: CRD42020190077) and followed PRISMA guideline. The manuscript is written very clearly and strictly. All sections are described in a comprehensive manner. Methods are adequate. I am very impressed with the work done.

Author’s Response: Thank you for your positive feedback. 

We have thoroughly proofread the manuscript for English language and spellcheck.